# Scc2/Nipbl hops between chromosomal cohesin rings after loading

James Rhodes[1], Davide Mazza[2,3], Kim Nasmyth[1]*, Stephan Uphoff[1]*

[1]Department of Biochemistry, Oxford University, Oxford, United Kingdom; [2]Istituto Scientifico Ospedale San Raffaele, Centro di Imaging Sperimentale, Milano, Italy; [3]Fondazione CEN, European Center for Nanomedicine, Milano, Italy

**Abstract** The cohesin complex mediates DNA-DNA interactions both between (sister chromatid cohesion) and within chromosomes (DNA looping). It has been suggested that intra-chromosome loops are generated by extrusion of DNAs through the lumen of cohesin's ring. Scc2 (Nipbl) stimulates cohesin's ABC-like ATPase and is essential for loading cohesin onto chromosomes. However, it is possible that the stimulation of cohesin's ATPase by Scc2 also has a post-loading function, for example driving loop extrusion. Using fluorescence recovery after photobleaching (FRAP) and single-molecule tracking in human cells, we show that Scc2 binds dynamically to chromatin, principally through an association with cohesin. Scc2's movement within chromatin is consistent with a 'stop-and-go' or 'hopping' motion. We suggest that a low diffusion coefficient, a low stoichiometry relative to cohesin, and a high affinity for chromosomal cohesin enables Scc2 to move rapidly from one chromosomal cohesin complex to another, performing a function distinct from loading.

DOI: https://doi.org/10.7554/eLife.30000.001

*For correspondence:
kim.nasmyth@bioch.ox.ac.uk (KN);
stephan.uphoff@bioch.ox.ac.uk
(SU)

**Competing interests:** The authors declare that no competing interests exist.

## Introduction

The organisation of chromosomes during interphase has an important role in the regulation of gene expression. Distal regulatory elements such as enhancers must be brought into proximity with their target promoters and shielded from inappropriate ones (insulation) (*Bulger and Groudine, 2010*). Recent advances in mapping DNA interactions have demonstrated that the human genome is organised into a series of sub-megabase, self-interacting regions called topologically associating domains (TADs) whose boundaries correspond to binding sites for the CCCTC-binding factor (CTCF) (*Nora et al., 2012*).

There is mounting evidence that the mechanism by which TADs and enhancer-promoter interactions are formed involves cohesin (*Kagey et al., 2010*; *Rollins et al., 1999*; *Wendt et al., 2008*). This Smc/kleisin complex holds sister chromatids together from their replication until chromosome segregation in mitosis (*Guacci et al., 1997*; *Michaelis et al., 1997*). The core cohesin complex is a ring-shaped heterotrimer of Smc1, Smc3 and Scc1 (Rad21) subunits (*Tóth et al., 1999*). Dimerization via their hinge domains creates V-shaped Smc1/Smc3 heterodimers whose apical head domains come together to form a composite ABC-like ATPase (*Haering et al., 2002*). Scc1's N-terminal domain binds to the Smc3 neck and its C-terminal domain to the Smc1 head thereby creating a closed ring (*Figure 1a*) (*Gruber et al., 2003*). It has been suggested that cohesin associates with chromatin by entrapping DNA within the ring's lumen while sister chromatid cohesion is mediated by co-entrapment of sister DNAs (*Haering et al., 2008*).

Cohesin's association with chromatin is regulated by several proteins among them the HAWKs (HEAT repeat containing proteins Associated With Kleisins): Scc2 (Nipbl), Pds5 and Scc3 (SA1/2) (*Wells et al., 2017*). Initial association of cohesin with DNA is regulated by Scc2 (*Ciosk et al., 2000*) and requires ATP hydrolysis (*Arumugam et al., 2003*). Scc2 is a large (316 kDa) hook-shaped protein

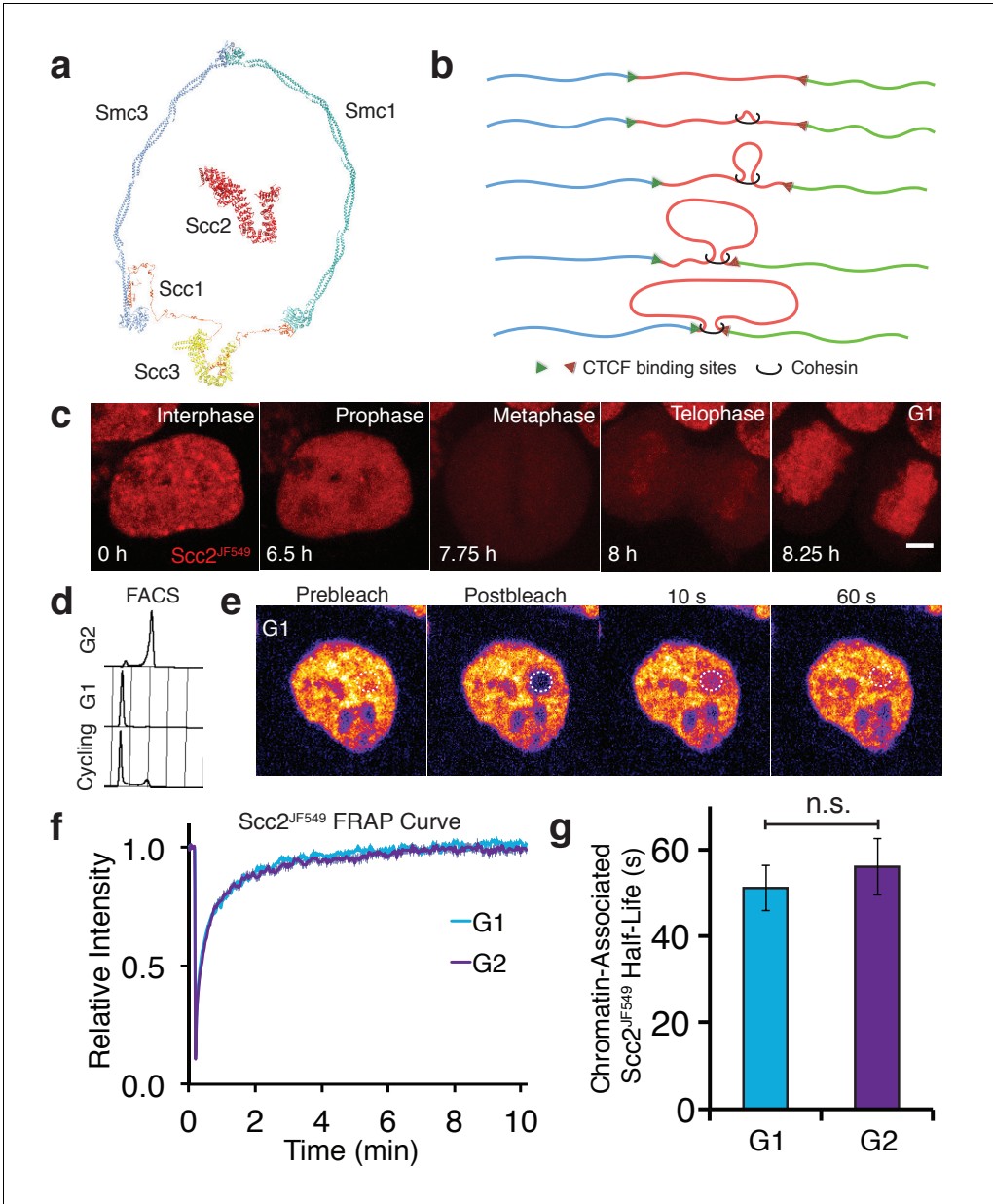

**Figure 1.** Scc2 interacts with chromatin independent of the cohesin loading reaction. (**a**) Model of the cohesin complex and Scc2 based on crystal structures (Scc2 (PDB: 5T8V), Smc1-Scc1 interface (PDB: 1W1W), Smc3-Scc1 interface (PDB: 4U × 3), Smc3-Smc1 interface (PDB: 2WD5), Scc3 (PDB: 4PJU), coiled coil of Smc3 and Smc1 modelled on dynein (PDB: 3WUQ), Scc1 central domain modelled from Scc2 N terminus (PDB: 4XDN). (**b**) Illustration to demonstrate the formation of TADs by loop extrusion. Loops are progressively enlarged until they reach convergent CTCF sites. (**c**) Z-projected images from a time-lapse confocal microscopy recording of JF549-Halo-Scc2 (Scc2$^{JF549}$) in HeLa cells. Time 0 hr = interphase, 6.5 hr = prophase, 7.75 hr = metaphase, 8 hr = telophase and 8.25 hr = G1. Scale bar = 5 μm. (**d**) FACS analysis of cells stained with propidium iodide either 6 hr (G2) or 15 hr (G1) after release from a double thymidine block, and cycling cells. (**e**) Still images from a fluorescence recovery after photobleaching (FRAP) experiment. Dashed circle represents bleached region. (**f**) FRAP curves of Scc2$^{JF549}$ in G1 and G2. Error bars denote standard error of the mean (s.e.m.). (**g**) Mean half-life of chromatin bound Scc2$^{JF549}$ derived from bi-exponential curve fitting of individual experiments from cells in G1 and G2. Error bars denote s.e.m. Unpaired t-test was used to compare conditions. n = 14 cells per condition.
DOI: https://doi.org/10.7554/eLife.30000.002

The following figure supplements are available for figure 1:

**Figure supplement 1.** Curve fitting of FRAP experiments.

*Figure 1 continued on next page*

*Figure 1 continued*

DOI: https://doi.org/10.7554/eLife.30000.003
**Figure supplement 2.** Curve fitting of FRAP experiments.
DOI: https://doi.org/10.7554/eLife.30000.004

whose N-terminal domain binds Scc4 (Mau2) to form the cohesin loading complex (*Kikuchi et al., 2016*). Though essential for loading cohesin onto yeast chromosomes, Scc2 is not required to maintain sister chromatid cohesion (*Ciosk et al., 2000*). Cohesin is released from DNA by Pds5 and Wapl (*Kueng et al., 2006*), which open the complex's Smc3-Scc1 interface (*Chan et al., 2012*). An equilibrium between loading and release gives cohesin rings a mean chromosome residence time of 15–30 min (*Gerlich et al., 2006*; *Hansen et al., 2017*).

It has been suggested that cohesin has the ability to extrude loops of DNA in a processive manner and that this process is halted by CTCF bound in one but not the other orientation (*Figure 1b*) (*Alipour and Marko, 2012*; *Fudenberg et al., 2016*; *Nasmyth, 2001*; *Sanborn et al., 2015*). If CTCF-regulated loop extrusion is responsible for TADs, then cohesin must be capable of translocating vast distances along chromatin fibres. Experiments in bacteria, yeast and mammalian cells and in vitro indicate that cohesin and its relatives have the ability to travel along DNA (*Busslinger et al., 2017*; *Davidson et al., 2016*; *Hu et al., 2011*; *Lengronne et al., 2004*; *Stigler et al., 2016*; *Wang et al., 2017*). A recent study has shown that cohesin's close relative condensin can translocate unidirectionallly along DNA and this motor activity depends on ATP hydrolysis (*Terakawa et al., 2017*). Scc2 stimulates cohesin's ATPase (*Murayama and Uhlmann, 2014*) and may play a role in the formation of TADs (*Haarhuis et al., 2017*). If cohesin's ATPase is required for its translocation along chromatin, then Scc2 might be expected to associate also with cohesin rings that have already loaded onto chromosomes. This has hitherto been addressed by ChIP sequencing (ChIP-Seq) studies, which have yielded conflicting results. Some found that Scc2 peaks overlap with those of cohesin only at enhancers and promoters (*Fournier et al., 2016*; *Kagey et al., 2010*). Others found little or no overlap with cohesin and instead detected Scc2 bound to active promoters that do not coincide with cohesin peaks (*Busslinger et al., 2017*; *van den Berg et al., 2017*; *Zuin et al., 2014*). Despite these differences none of the studies reported any significant co-localisation with CTCF sites where the vast majority of cohesin peaks are found. These discrepancies may be due to problems with crosslinking (*Teves et al., 2016*) or due to unreliable antibodies. Besides which, previous ChIP-Seq analyses suffer from a lack of calibration (*Hu et al., 2015*), which is necessary to distinguish genuine association from background noise.

Cornelia de Lange syndrome (CdLS) is a severe developmental disorder in which 60% of cases have heterozygous mutations in Scc2 (*Rohatgi et al., 2010*). However, cells from patients and a heterozygous Scc2 mouse model only display modest reductions in Scc2 expression (*Borck et al., 2006*; *Kawauchi et al., 2009*). It is not known why slightly reduced Scc2 abundance results in such severe developmental defects, but the level of cohesin on chromatin is unchanged and cohesion is unaffected in heterozygous Scc2 mice (*Chien et al., 2011*; *Remeseiro et al., 2013*). A cohesin-independent function in transcription has been suggested for Scc2 (*van den Berg et al., 2017*; *Zuin et al., 2014*) but further CdLS mutations are found in cohesin genes indicating an aetiology related to the complex (*Boyle et al., 2017*; *Deardorff et al., 2012*; *Revenkova et al., 2009*). These findings suggest that CdLS is not caused by a reduction in binding of cohesin to DNA but rather a change in its behaviour once loaded. However, no interaction has been demonstrated between Scc2 and loaded cohesin in vivo.

To determine whether Scc2 interacts with cohesin outside of the loading reaction we turned to live cell imaging. Using fluorescence recovery after photobleaching (FRAP) and single-molecule imaging, we show that Scc2 binds dynamically to chromatin, principally through an association with cohesin. In cells lacking Wapl, cohesin never dissociates from chromatin and accumulates along longitudinal axes called vermicelli (*Tedeschi et al., 2013*). We find that Scc2 co-localises with these axes but unlike cohesin, it turns over with a half-life of approximately one minute. Crucially, a pool of Scc2 with similar kinetics in wild type cells is greatly reduced after degradation of cohesin. This implies that a large fraction of chromosomal Scc2 is bound to cohesin at any moment in time. Scc2's movement within chromatin is consistent with a 'stop-and-go' or 'hopping' motion. We suggest that a low diffusion coefficient, a low stoichiometry relative to cohesin, and a high affinity for

chromosomal cohesin enables Scc2 to move rapidly from one chromosomal cohesin complex to another in its vicinity, performing a function distinct from loading.

## Results

### Scc2 interacts transiently with chromosomes before and after DNA replication

Because Scc2 is not stably associated with chromosomal cohesin (*Hu et al., 2011*), it has hitherto been assumed to only interact with cohesin transiently during the loading process. For this reason as well as the difficulties of ascertaining the location of proteins with short chromosome residence times using ChIP-Seq, we investigated Scc2's dynamics using live-cell imaging. To do this, we tagged Scc2 at its N-terminus with the HaloTag in HeLa cells using CRISPR/Cas9-mediated homologous recombination (*Stewart-Ornstein and Lahav, 2016*). Halo-Scc2 was labelled by transient incubation with the fluorescent dye JF549 conjugated to the HaloTag ligand (Scc2$^{JF549}$) (*Grimm et al., 2015*). Timelapse confocal microscopy confirmed previous findings from immunofluorescence experiments (*Watrin et al., 2006*), that Scc2 is nuclear during interphase, dissociates from chromatin in prophase and is excluded from chromosomes during mitosis (*Figure 1c*, *Video 1*).

To compare Scc2's dynamics at different stages of the cell cycle, we obtained G1 and G2 populations by releasing cells for different periods of time from a double thymidine block. Cells were predominantly in G2 6 hr after release, having just completed S phase, while they were predominantly in G1 15 hr after release, having undergone both DNA replication and mitosis (*Figure 1d*). The interaction between Scc2 and chromatin was measured by bleaching a circle of Scc2$^{JF549}$ fluorescence and measuring fluorescence recovery after photobleaching (FRAP) (*Figure 1e,f*). FRAP experiments were performed in the presence of an unlabelled HaloTag ligand to prevent relabeling of newly synthesised Halo-Scc2 (*Rhodes et al., 2017*). Scc2$^{JF549}$ FRAP curves did not fit a single exponential function (*Figure 1—figure supplement 1*). However, a double exponential model fitted the recovery data from both sets of cells (*Figure 1—figure supplements 1* and *2*). In G1 cells, 53% of the fluorescence recovered with a half-life of 2.9 s and 45% with a half-life of 51 s, while in G2 cells, 57% of the fluorescence recovered with a half-life of 3.9 s and 41% with a half-life of 56 s (*Figure 1g*). Our results indicate that Scc2's association with chromatin is much more transient and frequent than that of cohesin which has a residence time of 15–30 min (*Gerlich et al., 2006*; *Hansen et al., 2017*). During DNA replication 30% of cohesin becomes stably bound on chromosomes with a residence time in the hours range (*Gerlich et al., 2006*). Unlike cohesin, there are only modest differences in Scc2 recovery characteristics between G1 and G2 cells.

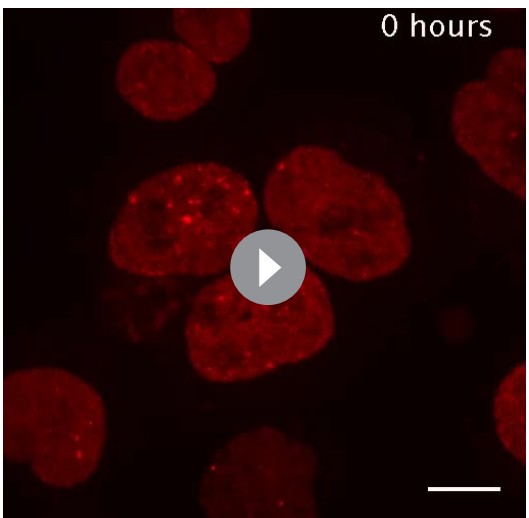

**Video 1.** Time-lapse live cell microscopy of Scc2$^{JF549}$ in wild type HeLa cells. Scale bar = 5 μm.
DOI: https://doi.org/10.7554/eLife.30000.005

### Scc2 co-localises with cohesin vermicelli in Wapl Δ cells

Because cohesin and Scc2 appear evenly distributed within nuclei by conventional fluorescence microscopy, it is hard to address whether they co-localise. Deletion of the cohesin release factor Wapl results in re-organisation of cohesin into axial structures called vermicelli (*Tedeschi et al., 2013*). We used this phenomenon to determine whether Scc2 co-localises with chromosomal cohesin. To this end, we transfected Halo-Scc2 HeLa cells with a plasmid expressing Cas9 and a guide RNA that together make a double strand break in *WAPL's* M1116 codon, a residue essential for Wapl's releasing activity (*Ouyang et al., 2013*). This causes deletions in most genes but also gives rise to M1116 mutations (*Rhodes et al., 2017*). Three days post transfection, immunofluorescence with an antibody

against Scc1 showed that cohesin had re-organised into vermicelli in most cells (*Figure 2a*). Strikingly, Scc2[JF549] largely co-localised with the cohesin vermicelli and not with the majority of DNA that surrounds these structures (*Figure 2a*). However, in contrast to cohesin, which is permanently associated with chromosomes in WaplΔ mutants (*Tedeschi et al., 2013*), Scc2[JF549] still showed fast FRAP recovery after Wapl inactivation (*Figure 2b, c and d*, *Figure 2—figure supplement 1*, *Video 2*).

Upon inactivation of Wapl the fraction of cohesin associated with chromatin increases (*Kueng et al., 2006*) and the unbound fraction is reduced. In this situation, one would expect the frequency of Scc2's association with chromatin to decrease if Scc2 and cohesin interacted only during the cohesin loading reaction. This is because there are fewer cohesin complexes available for an Scc2-mediated loading reaction. In fact, we observed precisely the opposite. The fraction of Scc2 bound to chromatin increased when Wapl was inactivated (*Figure 2c*), despite less unbound cohesin being available to load onto DNA. Increased chromatin binding of Scc2 therefore appears to reflect an association between Scc2 and cohesin that is stably loaded on DNA. Scc2's continual albeit transient association with vermicelli may regulate aspects of cohesin function besides loading.

Analysis of the Wapl defective cells was also revealing about Scc2's behaviour during mitosis. Most cohesin dissociates from chromosome arms when cells enter M phase (*Losada et al., 1998*). This process, which is known as the prophase pathway, involves the same mechanism responsible for cohesin's turnover during interphase, namely Wapl-mediated opening of the ring's Smc3/Scc1 interface (*Chan et al., 2012*). In cells lacking Wapl, cohesin persists throughout chromosomes until separase removes it during anaphase (*Kueng et al., 2006*) (*Figure 2e*). This situation presents an opportunity to address whether the lack of Scc2's association with chromosomes from prophase till metaphase is simply due to the lack of cohesin or due to cell cycle regulation of Scc2's ability to bind cohesin. In other words, does Scc2 still dissociate from chromosomes in WaplΔ cells during mitosis? Time-lapse and immunofluorescence microscopy of Scc2[JF549] WaplΔ HeLa cells demonstrated that Scc2 dissociates from chromosomes in prophase even in the absence of Wapl activity (*Figure 2e* and *Video 3*). Thus, the prophase release of Scc2 is independent of cohesin release. Activation of mitotic protein kinases during prophase may abrogate Scc2's ability to bind to chromosomal cohesin.

## Scc2 hops along the vermicelli of WaplΔ cells

Observation of fluorescence recovery after photobleaching a large fraction of the nucleus revealed a striking phenomenon. Given Scc2's rapid turnover on chromatin, one would expect Scc2 molecules that have dissociated from chromatin to reappear rapidly throughout the bleached zone, as is the case in most FRAP studies on proteins with short chromosome residence times. Surprisingly, Scc2[JF549] behaved very differently. Upon photobleaching one half of a nucleus, fluorescence associated with Scc2[JF549] spread into the bleached zone very slowly, taking longer than five minutes to equilibrate in zones furthest from the unbleached area (*Figure 3a and b*). This implies that Scc2's diffusion through the nucleus is severely restricted. One explanation for this low mobility is that Scc2 diffuses extremely slowly through the nucleoplasm. Alternatively, soluble Scc2 may rebind chromatin before it diffuses appreciably. In other words, its diffusion is continually punctuated by re-binding and re-dissociation.

In wild type cells it is difficult to distinguish between these two possibilities, as Scc2 is homogeneously distributed. To differentiate between DNA-bound and unbound Scc2, we used Wapl deficient cells where bound Scc2 forms vermicelli. After photobleaching one half of the nucleus where Scc2[JF549] was associated with the cohesin vermicelli, we observed that fluorescence spread in a gradual fashion into the bleached zone and associated with vermicelli as it did so (*Figure 3c*). Fluorescence appeared earliest on those vermicelli closest to the unbleached zone and latest on those furthest from the unbleached zone. In other words, the movement of Scc2[JF549] across the nucleus took place while it was continually associating with and dissociating from vermicelli. Thus, upon dissociation from one cohesin complex, Scc2 rebinds a neighbouring one before it can diffuse an appreciable distance across the nucleus. It appears therefore to 'hop' across the nucleus on chromosomal cohesin. Similar hopping behaviour has been suggested to occur for the histone linker H1 and a class of pioneering transcription factors (*Misteli et al., 2000*; *Sekiya et al., 2009*).

To confirm that this behaviour was not an artefact caused by the HaloTag, we repeated the experiment in HeLa cells expressing a mouse GFP-Scc2 under its endogenous promoter from a

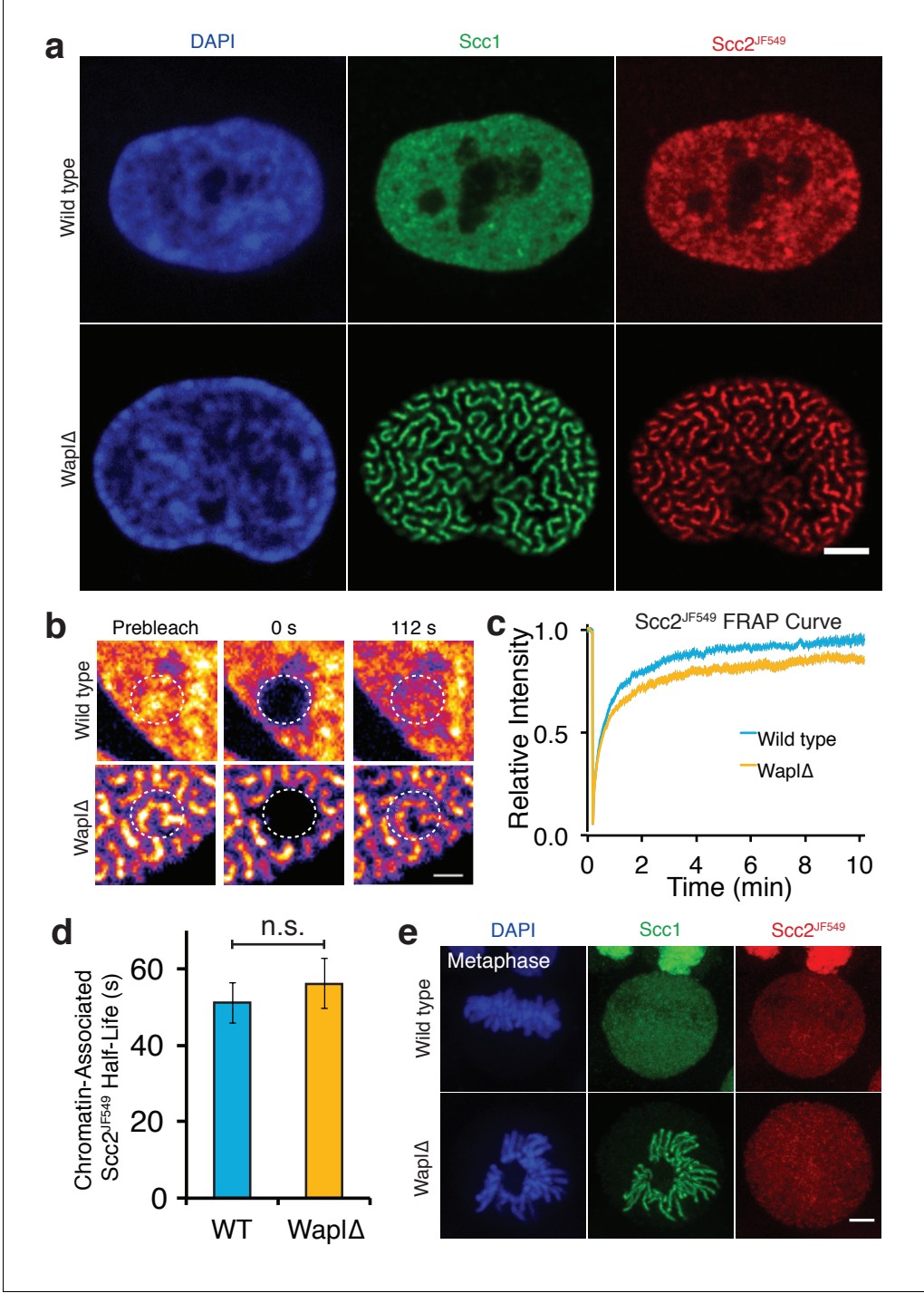

**Figure 2.** Scc2 binds to cohesin that is already loaded on DNA. (a) Immunofluorescence microscopy images of wild type and WaplΔ Halo-Scc2 HeLa cells. Cohesin was stained with an antibody against Scc1 and Halo-Scc2 with JF549. Scale bar = 5 μm. (b) Still images from spot FRAP experiments on Scc2$^{JF549}$ in asynchronous wild type or WaplΔ HeLa cells. Dashed circle represents bleached region. Scale bar = 1 μm. (c) FRAP recovery curves from wild type and WaplΔ cells. Error bars denote s.e.m. n = 14 cells per condition. (d) Mean half-life of chromatin bound Scc2$^{JF549}$ derived from bi-exponential curve fitting of individual experiments from wild type or WaplΔ cells. Error bars denote s.e.m. Unpaired t-test was used to compare conditions. n = 14 cells per condition. (e) Immunofluorescence microscopy images of wild type or WaplΔ Halo-Scc2 HeLa cells in metaphase. Cells were stained as in a. Scale bar = 5 μm.

DOI: https://doi.org/10.7554/eLife.30000.006

*Figure 2 continued on next page*

*Figure 2 continued*
The following figure supplement is available for figure 2:
**Figure supplement 1.** Curve fitting of FRAP experiments.
DOI: https://doi.org/10.7554/eLife.30000.007

stably integrated bacterial artificial chromosome (BAC). Again we observed gradual spreading from unbleached into bleached zones along vermicelli (*Figure 3—figure supplement 1* and *Video 4*).

## Scc2's chromosomal association depends on cohesin

If an appreciable fraction of chromatin bound Scc2 is indeed associated with cohesin, then Scc2's dynamics should be greatly altered by removing cohesin from the cell. Because effective cohesin depletion will have major ramifications on cell cycle progression, which in itself would affect Scc2's dynamics, it is essential to measure the effect in cells in which cohesin has been depleted extremely rapidly and before cells enter mitosis. To this end, we used an HCT116 human cell line whose cohesin subunit Scc1 is tagged with an auxin-inducible degron (mAID) and a fluorescent mClover tag. These cells also express the plant F-box protein Tir1 that mediates interaction of the AID degron with endogenous SCF ubiquitin ligase (*Natsume et al., 2016*).

To measure Scc2's dynamics in these cells before and after auxin-mediated Scc1 degradation, we again used CRISPR to tag Scc2 at its N-terminus with the HaloTag. Addition of auxin induced degradation of Scc1 to levels below detection by microscopy within two hours (*Figure 4a and b*). To compare the dynamics of Scc2 with those of a protein of similar size, we created a second HCT116 cell line in which both *SCC1* genes were tagged with the HaloTag. The molecular weight of the Smc1, Smc3, Scc1, Scc3 tetramers is 500 kDa while that of Scc2/Scc4 is 386 kDa. Importantly, 50% of cohesin is not bound to chromatin in interphase cells and known to diffuse freely within the nucleoplasm due to a low association rate (*Hansen et al., 2017*).

We initially analysed Scc2[JF549] FRAP within nuclei in which one half had been photobleached. FRAP of Scc2[JF549] in Scc1-mAID-mClover Tir1 cells in the absence of auxin revealed slow spreading of Scc2[JF549] into the unbleached half of the nucleus, as previously found in HeLa cells. Crucially, recovery of Scc2[JF549] was much slower than that of the freely diffusing pool of Scc1[JF549], confirming that Scc2's diffusion through the nucleus is an interrupted process, and not simply a consequence of its high molecular weight (*Figure 4c*). Addition of auxin caused complete depletion of Scc1 within two hours, as measured by mClover fluorescence intensity (*Figure 4b*). Strikingly, this was accompanied by a major increase in the rate of Scc2[JF549] fluorescence recovery after photobleaching (*Figure 4c*, *Figure 4—figure supplement 1*). It is conceivable that the increase in the rate of recovery upon Scc1 degradation is due to an interaction between Scc2 and the soluble pool of cohesin, which could somehow slow diffusion in the nucleoplasm. Because the diffusion coefficient of unbound Scc2 molecules was in fact unchanged by the presence or absence of cohesin in the cell (see below), we conclude that it is chromosomal cohesin and not the soluble pool that hinders Scc2's diffusion.

These data imply that Scc2's slow movement through the nucleus is due to it hopping between neighbouring chromosomal cohesin complexes. Importantly, the behaviour of Scc2 in WaplΔ cells shows that it binds and then rapidly dissociates from cohesin complexes that are themselves permanently locked onto chromosomes. In other words, Scc2 does not merely bind to cohesin during the loading process.

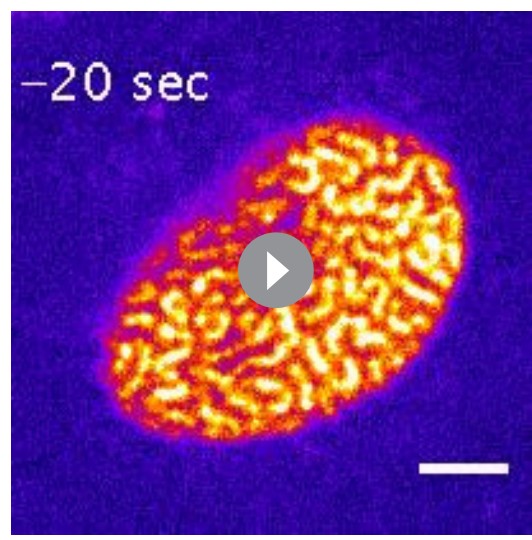

**Video 2.** Spot FRAP of Scc2[JF549] in WaplΔ HeLa cells. Scale bar = 5 µm.
DOI: https://doi.org/10.7554/eLife.30000.008

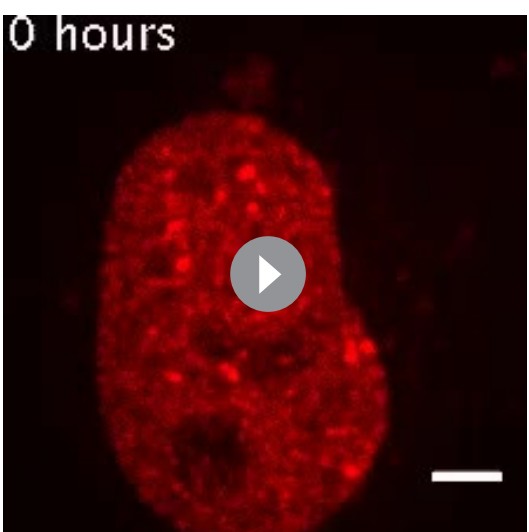

**Video 3.** Time-lapse live cell microscopy of Scc2[JF549] in WaplΔ HeLa cells. Scale bar = 5 μm.
DOI: https://doi.org/10.7554/eLife.30000.009

Given that Scc2 is a potent activator of hydrolysis of ATP by cohesin (*Murayama and Uhlmann, 2014*), our discovery that Scc2 cycles on and off chromosomal cohesin raises the possibility that it stimulates ATP hydrolysis by chromosomal cohesin complexes not just ones engaged in loading.

To address whether Scc2 can bind to chromatin even in the absence cohesin, we repeated the FRAP experiment in Scc1-depleted HCT116 cells, but in this case photobleaching just a small circular area (*Figure 4d*), as described for *Figures 1c* and *2b*, which enabled us to model the recovery curves. These were inconsistent with a single exponential function but fitted a bi-exponential function well, indicating that Scc2 interacts with chromatin even in the absence of cohesin. The lack of cohesin simplified Scc2's dynamics and enabled us to calculate a residence time from the FRAP curves (*Mueller et al., 2008*). Model fitting revealed that in cohesin-depleted HCT116 cells 44% of Scc2 binds to chromatin with a residence time of 22 s. The unbound fraction moves with a diffusion coefficient of 0.79 μm²/s. This slow diffusion coefficient might indicate that even in the absence of cohesin, Scc2 moves by effective diffusion (where diffusion is interrupted by transient binding). Our findings imply that Scc2 binds to chromosomes in two modes: one involving cohesin and a second more transient one to other chromatin sites, potentially reflecting the previously reported association with gene promoters (*Zuin et al., 2014*).

## Single-molecule imaging demonstrates Scc2 binding to cohesin in wild type cells

The FRAP measurements suggest that the association between Scc2 and cohesin has a high on rate as well as a high off rate and that Scc2 may also have a relatively low diffusion coefficient within the nucleoplasm. It would also seem that while at a given moment there might be a significant unbound fraction of Scc2 in the nucleus, this protein cannot diffuse very far as its movement is interrupted by frequent binding events. To test these predictions, we employed single-molecule imaging to visualise directly the movement of Scc2 molecules and quantify their interactions. The Halo ligand JF549 is sufficiently bright to detect single Halo-Scc2 molecules at 15 ms exposures and ~25 nm localisation precision inside nuclei of live HCT116 cells (*Figure 5a*). As previously demonstrated (*Liu et al., 2014*), the JF549 dye blinks stochastically, allowing sequential imaging and localisation of thousands of molecules per cell over the course of a movie. Single molecules were visible for an average of 9 frames (135 ms) before blinking, photobleaching or moving out of the focal plane. In some cases, molecules were visible for several seconds. By linking localisations to tracks, we constructed maps of Scc2 movement inside nuclei, where the colour of each track represents the average apparent diffusion coefficient per molecule (*Figure 5b*). This analysis revealed immobile Scc2 molecules (blue-cyan tracks) as well as molecules displaying clear displacements between successive frames (yellow-red tracks) (*Figure 5c*). The distribution of diffusion coefficients revealed two distinct populations: 37% displayed a diffusion coefficient compatible with chromatin bound molecules while 63% were mobile and therefore unbound (*Figure 5d and e*). The average apparent diffusion coefficient of the unbound molecules was 0.6 μm²/s, consistent with our results from FRAP.

To compare Scc2's movement with that of its binding partner cohesin, we performed identical tracking experiments in HCT116 cells where Scc1-Halo was labelled with JF549. The distribution of diffusion coefficients for Scc1 was remarkably similar to Scc2, showing distinct subpopulations of bound and diffusing molecules (*Figure 5d and e*), as reported previously (*Hansen et al., 2017*). Tagging Scc1 with Halo also enabled us to compare the stoichiometry of Scc2 and Scc1 proteins. Fluorescence associated with Scc1[JF549] was nearly three times that associated with Scc2[JF549] (*Figure 5f*),

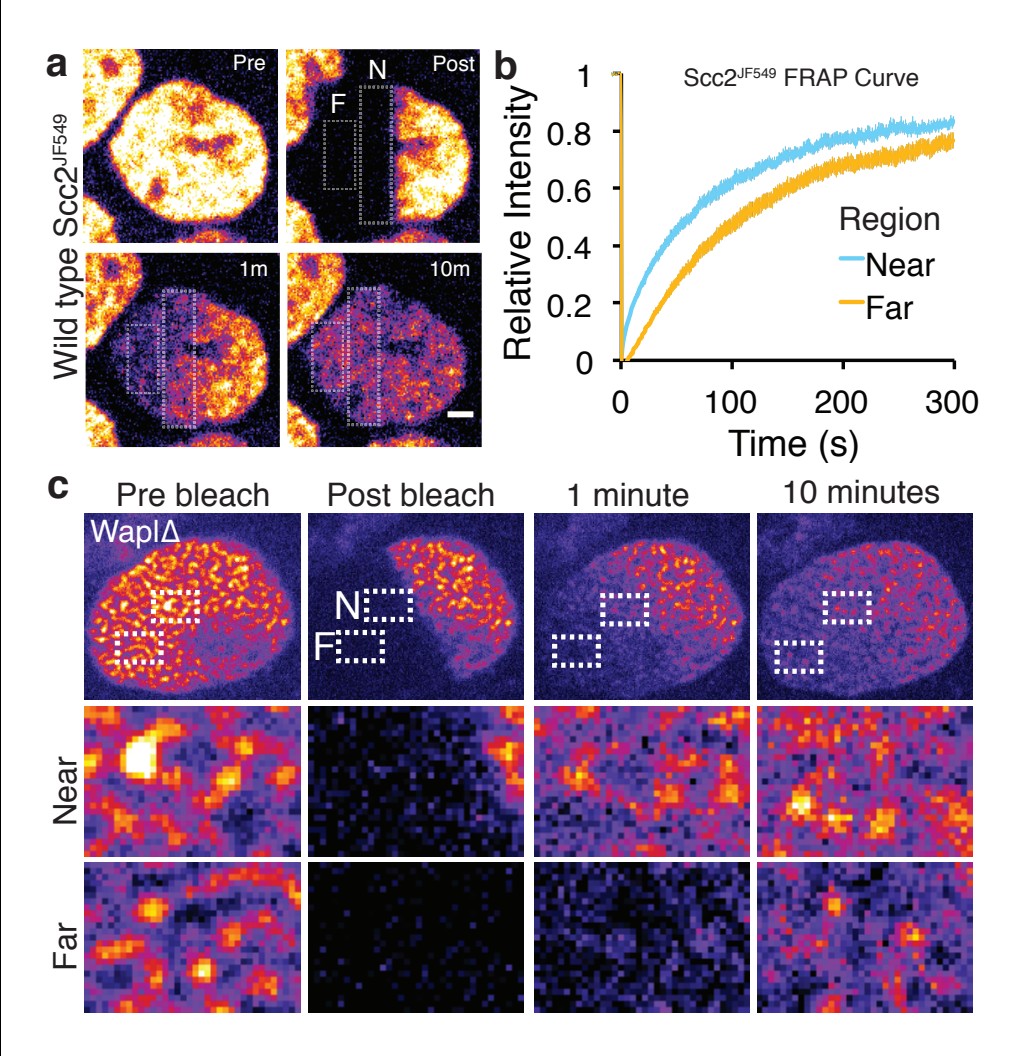

**Figure 3.** Scc2 hops on chromatin. (a) Stills from half nuclear FRAP of Scc2$^{JF549}$ in wild type HeLa cells. Dashed rectangle highlight a region Near (N) to and a region Far (F) from the unbleached half. Scale bar = 2.5 μm. (b) Half-nuclear FRAP curves of Scc2$^{JF549}$ in wild type HeLa cells. Recovery curves are shown from two zones within the bleached region. One zone is Near to the unbleached zone and the other is Far from the unbleached zone. Error bars denote s.e.m. n = 14 cells per condition. (c) Still images from a half-nuclear FRAP experiment of Scc2$^{JF549}$ in WaplΔ HeLa cells. Dashed rectangle highlight a zone near (N) and a zone Far (F) from the unbleached region shown in insets. Scale bar = 1 μm in inset.

DOI: https://doi.org/10.7554/eLife.30000.010

The following figure supplement is available for figure 3:

**Figure supplement 1.** Scc2 hops on chromatin.

DOI: https://doi.org/10.7554/eLife.30000.011

while the relative fractions of chromatin-bound molecules were the same for Scc1 and Scc2 (*Figure 5e*). Therefore, Scc2 is present at a substoichiometric level relative to DNA-bound cohesin. Hence there is an abundance of binding sites for Scc2, which may contribute to the gradual spreading of fluorescence, observed by FRAP.

As a direct test for our model that Scc2 repeatedly binds pre-loaded cohesin, we tracked Scc2 in cells in which the abundance of chromatin-bound cohesin had been perturbed. First, we employed Wapl deficient HeLa cells where most cohesin is chromatin-bound and found that the fraction of bound Scc2 molecules increased from 41 ± 3% (wild type) to 55 ± 1% (WaplΔ) (*Figure 6a*). Thus, increasing the abundance of DNA-bound cohesin leads to greater recruitment of Scc2. Next, we

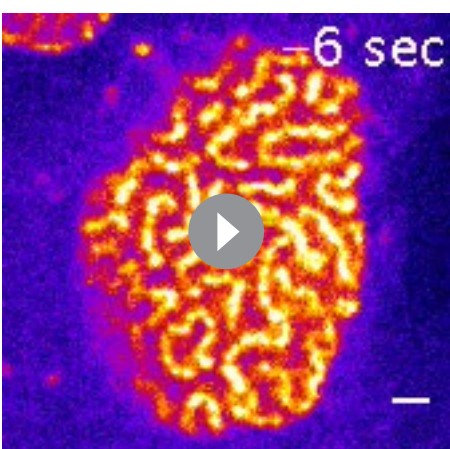

**Video 4.** Stripe FRAPof GFP-Scc2 in WaplΔ HeLa cells. Scale bar = 2 μm.
DOI: https://doi.org/10.7554/eLife.30000.012

analysed the effect of depleting cohesin using auxin-mediated Scc1 degradation in HCT116 cells. As expected, this had the opposite effect, namely that Scc2 binding decreased from 37 ± 2% (untreated) to 27 ± 2% (with auxin) (*Figure 6b*). These findings are fully consistent with the notion that a sizeable fraction of Scc2 is bound directly to cohesin at any one time. The fact that a significant fraction of immobile Scc2 molecules remains after cohesin degradation merely confirms that Scc2 is capable of binding sites independently of cohesin. Because of this, the decrease in the fraction of immobile Scc2 upon cohesin depletion will not in fact reflect the fraction of Scc2 that is normally associated with cohesin. This is better estimated from the effect of cohesin depletion on FRAP recovery curves (*Figure 4c*), which can distinguish the two types of chromosomal association (cohesin-dependent and independent) because they have different residence times.

Interestingly, although the relative abundance of unbound Scc2 molecules was reduced by Wapl deletion and increased by cohesin degradation, the apparent diffusion coefficients of unbound Scc2 molecules remained unchanged in both situations. This indicates that the movement of Scc2 during the search for binding sites is not affected by the presence or absence of cohesin or by the reorganisation and compaction of chromatin caused by Wapl deletion.

## Direct observation and quantification of Scc2-cohesin binding

Single-molecule tracking should enable direct observation of Scc2's transient binding events in wild type cells. We examined long-lived tracks of Scc2 and frequently observed instances where the diffusion coefficient changed during the trajectory. Single molecules displayed transient binding events that lasted a few hundred milliseconds followed by dissociation and intervals of diffusive motion (*Figure 7a*). This is direct evidence for the effective diffusion postulated from our FRAP experiments. However, we also found that many molecules remained immobile on a much longer time-scale. Furthermore, after degrading Scc1 by auxin treatment, Scc2 still displayed transient binding events as seen in untreated cells. Therefore, we interpret these binding events on a time-scale of ~100 ms as cohesin-independent chromatin interactions during the target search.

Measuring the binding times of the long-lived immobile species was complicated by photobleaching, which limits how long each molecule can be imaged. To capture long-lived binding events more efficiently, we imaged Scc2$^{JF549}$ in movies at a slower frame rate of 1 s/frame and lower laser intensity. Under these conditions, diffusing molecules are blurred and therefore not detected by the localisation analysis, whereas stationary or slowly moving molecules are detected as sharp diffraction-limited spots (*Figure 7b*) (*Mazza et al., 2012*; *Uphoff et al., 2013*). However, the observed dwell times are biased by loss of signal due to photobleaching, blinking, drift, and localisation errors. Therefore, we first calibrated the method by measuring the dwell time distribution of chromosomal Scc1, which is known to be stably bound for tens of minutes (*Gerlich et al., 2006*). Any loss of bound Scc1 molecules on a shorter time-scale than its known residence time must be due to the aforementioned experimental artefacts. We calculated this loss rate by fitting a double-exponential decay to the measured dwell time distribution of Scc1 and applied this correction factor to calculate binding time constants for Scc2. The dwell times showed a characteristic double exponential distribution (*Figure 7b*). 55% of Scc2 molecules were bound with a half-life of 1 s and 45% bound with a half-life of 47 s. These values are in reasonable agreement with those obtained from FRAP analysis (53% and 45% with half lives of 2.9 and 51 s, respectively).

Our single-molecule tracking experiments explain the slow spreading of Scc2 fluorescence seen in FRAP experiments. By resolving chromatin-bound and diffusing subpopulations of Scc2, we have shown in as direct a manner as possible that Scc2 associates with chromatin-bound cohesin

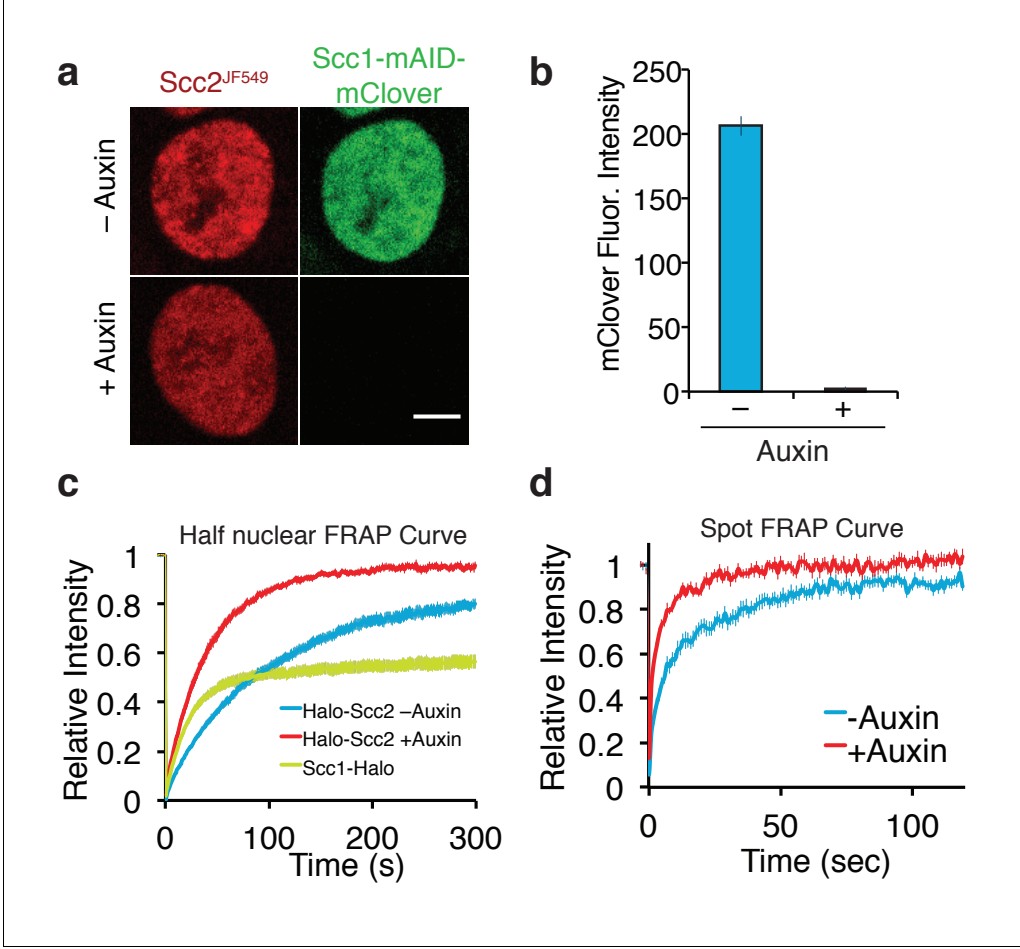

**Figure 4.** Depletion of core cohesin subunit Scc1 releases most, but not all, Scc2 from chromatin. (**a**) Live cell microscopy images of Scc1-mClover-mAID cells ± auxin (500 µM, 1h30 incubation). Scale bar = 5 µm. (**b**) Graph of fluorescence intensity of Scc1-mClover-mAID ± auxin demonstrates Scc1 degradation. n = 14 cells per condition. (**c**) Half-nuclear FRAP recovery curves of asynchronous HCT116 cells ± auxin. Error bars denote s.e.m. n = 14 cells per condition. (**d**) Spot FRAP recovery curves from asynchronous HCT116 cells ± auxin. Error bars denote s.e.m. n = 13 cells per condition.

DOI: https://doi.org/10.7554/eLife.30000.013

The following figure supplement is available for figure 4:

**Figure supplement 1.** Curve fitting of FRAP experiments.

DOI: https://doi.org/10.7554/eLife.30000.014

complexes where it has a residence time of approximately one minute. The high relative abundance of chromosomal cohesin allows Scc2 to rapidly bind a nearby complex after dissociation. Furthermore, in-between cohesin binding events, Scc2 displays effective diffusion where its movement is frequently halted by transient chromatin binding on a sub-second time-scale (*Figure 7c*). This allows Scc2 to diffuse locally in order to stay in close contact with cohesin complexes.

## Discussion

Given that Scc2 is the most frequently mutated protein in Cornelia de Lange syndrome it is critical to understand the nature of its interactions. Scc2 stimulates cohesin's ATPase (*Murayama and Uhlmann, 2014*) and may play a key role in translocating cohesin along chromatin fibres (*Kanke et al., 2016*), and the possible extrusion of DNA loops (*Haarhuis et al., 2017*). If this were true one would expect cohesin and Scc2 to reside at the same locations on chromosomes. However, co-localisation has not been observed between cohesin and Scc2 at CTCF sites where the majority of cohesin ChIP-

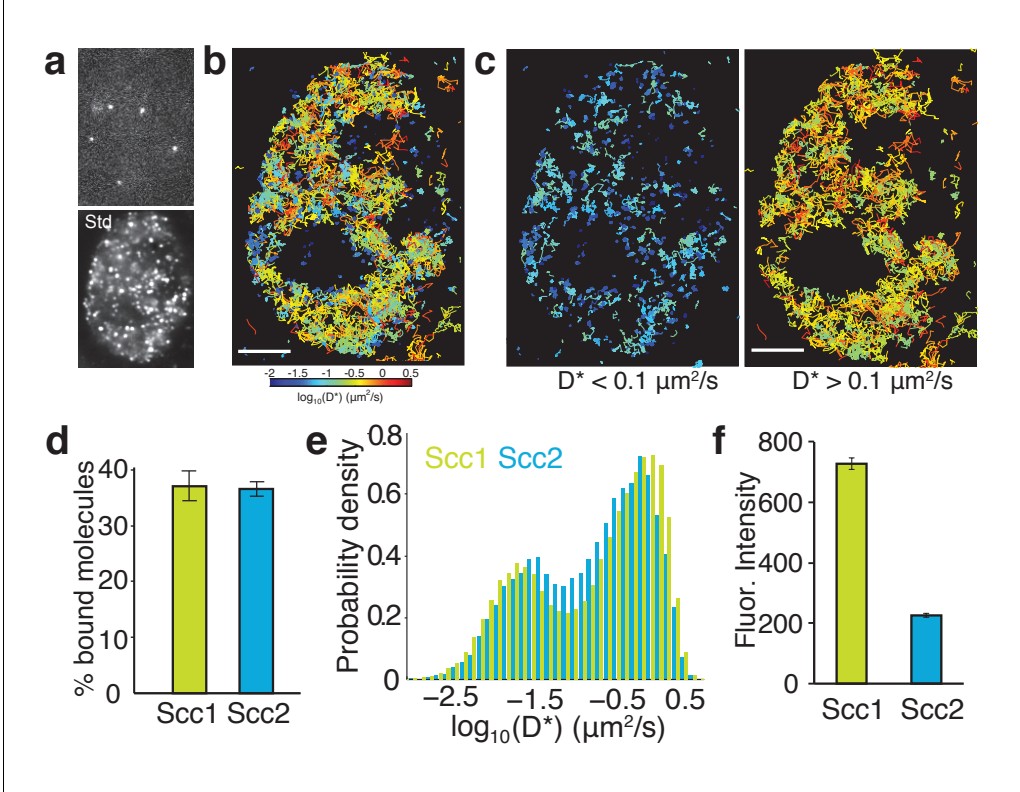

**Figure 5.** Single-molecule tracking of Scc2 and Scc1 in live cells. (**a**) Example frame from a tracking movie showing fluorescent spots of single Scc2[JF549] molecules. Standard deviation (Std) of pixel intensities from a movie shows the spatial distribution of Scc2[JF549]. (**b**) Map of Scc2[JF549] tracks in an HCT116 cell. Each track shows the movement of a single molecule; colours represent the average diffusion coefficient per track. (**c**) Tracks of immobile (D*<0.1 μm2/s) and mobile (D*>0.1 μm2/s) Scc2[JF549] molecules. (**d**) Percentage of molecules classified as immobile (D*<0.1 μm2/s) for Scc2 and Scc1. n > 10 cells. (**e**) Log-scale distribution of apparent diffusion coefficients D* for Scc2 (blue) and Scc1 (green). n > 10 cells. (**f**) Fluorescence intensity (a.u.) of Scc2[JF549] and Scc1[JF549]. n > 10 cells.
DOI: https://doi.org/10.7554/eLife.30000.015

Seq peaks are found. Here we applied a very different approach, namely imaging fluorescently labelled versions of Scc2 in living cells.

Our ability to resolve transient chromatin interactions enabled the surprising discovery that Scc2 frequently rebinds to cohesin complexes that have already been loaded onto DNA. Therefore, Scc2 appears to serve a function apart from its documented role as a cohesin loader. We found that Scc2 co-localises with cohesin along the longitudinal axes of interphase chromosomes observed in WaplΔ cells. Under these conditions, cohesin is stably bound to chromatin whereas Scc2 turned over with a half-life of approximately one minute. If Scc2 formed these vermicelli only because of chromatin rearrangement, the fraction of Scc2 bound to DNA should be unchanged. In fact, the abundance of chromatin-bound Scc2 increased after Wapl deletion. This excludes the possibility that the interaction between Scc2 and chromatin exists merely because of an association of Scc2 and cohesin during the initial loading reaction or an association with gene regulatory elements. Instead, the simplest explanation for this behaviour is that Scc2 binds transiently but continually to previously loaded cohesin complexes. As predicted by this hypothesis, acute cohesin depletion greatly increases Scc2's mobility within the nuclei of wild type cells. It also reduces the fraction of chromatin-bound molecules. This effect is more modest than the effect on mobility because Scc2 also binds to chromatin in the absence of cohesin, albeit with a considerably shorter residence time.

These findings beg the question what the function is that Scc2 plays when it binds to loaded cohesin. One possibility is to stimulate cohesin's translocation along chromatin fibres and thereby the extrusion of DNA loops. These processes might require ATPase activity associated with Smc1

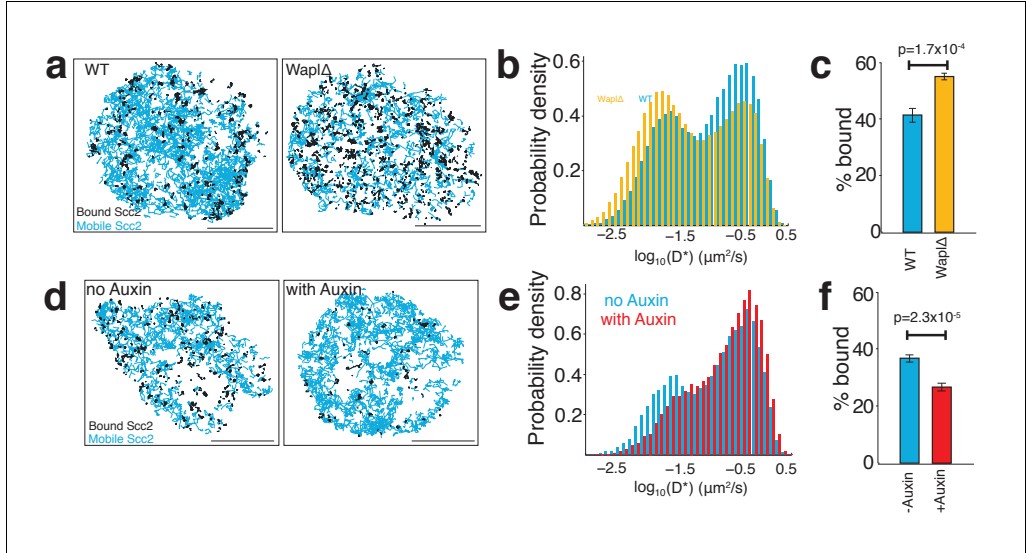

**Figure 6.** Scc2 binding is altered by the abundance of chromatin-associated cohesin. (a) Maps of Scc2[JF549] tracks in wild type and Wapl deficient HeLa cells with immobile molecules (D*<0.1 µm2/s) in black and mobile molecules (D*>0.1 µm2/s) in blue. Scale bars = 5 µm. (b) Log-scale distribution of apparent diffusion coefficients D* of Scc2 in wild type and Wapl deficient cells. n > 10 cells. (c) Percentage of immobile Scc2 molecules in wild type and Wapl deficient cells. Unpaired t-test was used to compare conditions. n > 10 cells. (d) Maps of Scc2[JF549] tracks in HCT116 cells ± auxin-mediated degradation of Scc1. Immobile molecules (D*<0.1 µm2/s) shown in black and mobile molecules (D*>0.1 µm2/s) in blue. n > 10 cells. (e) Log-scale distribution of apparent diffusion coefficients D* of Scc2 ± degradation of Scc1 with auxin. n > 10 cells. (f) Percentage of immobile Scc2 molecules ± degradation of Scc1 with auxin. Unpaired t-test was used to compare conditions. n > 10 cells.

DOI: https://doi.org/10.7554/eLife.30000.016

and Smc3 (*Kanke et al., 2016*), a reaction that is stimulated by Scc2 (*Murayama and Uhlmann, 2014*). Further experiments permitting manipulation of Scc2 activity will be required to address what function Scc2 performs on chromosomal cohesin. Ascertaining whether it facilitates loop extrusion and thereby formation of TADs will be an important goal.

It is intriguing that ChIP-Seq studies have missed Scc2's close association with chromosomal cohesin. We have already mentioned technical reasons why ChIP-Seq may not have revealed Scc2's real location. However, there may be an equally important reason. Because Scc2 and cohesin ChIP-Seq measurements have not been calibrated (*Hu et al., 2015*), their analyses have focused on local maxima, on the assumption that these must be genuine signals. We note that the majority of cohesin ChIP-Seq reads are in fact not situated in peaks but are instead distributed throughout the genome (*Landt et al., 2012*). If in fact these reads also represent genuine association, as suggested by calibrated ChIP-Seq in yeast (*Hu et al., 2015*), then by focusing solely on cohesin peaks, ChIP-Seq analyses may have grossly under-estimated Scc2's co-localisation with cohesin. Indeed, if Scc2 mediates the ATP hydrolysis necessary to drive loop extrusion, then one would predict that Scc2 would be associated with cohesin complexes that are engaged in extrusion, which Hi-C studies suggest are distributed broadly throughout the genome, and possibly not with those that have reached boundaries created by CTCF. Thus, the apparent lack of co-localisation between cohesin and Scc2 concluded by ChIP-Seq analyses may in fact be telling us something far more revealing. It is conceivable that CTCF may actively prevent cohesin's association with Scc2, stop ATP hydrolysis by cohesin and thereby halt extrusion of loops beyond CTCF binding sites. In other words, insulation might be mediated by an effect of CTCF on Scc2-driven ATP hydrolysis.

Cornelia de Lange Syndrome is caused by heterozygous mutations in *SCC2* in 60% cases (*Rohatgi et al., 2010*). Mouse models of the disorder with a heterozygous deletion of *SCC2* display severe defects but only have a 30% reduction in *SCC2* expression (*Kawauchi et al., 2009*). Consistently, a CdLS case has also been reported in which the patient displayed a clinically significant phenotype but only a 15% drop in *SCC2* mRNA expression due to a mutation in the 5' untranslated

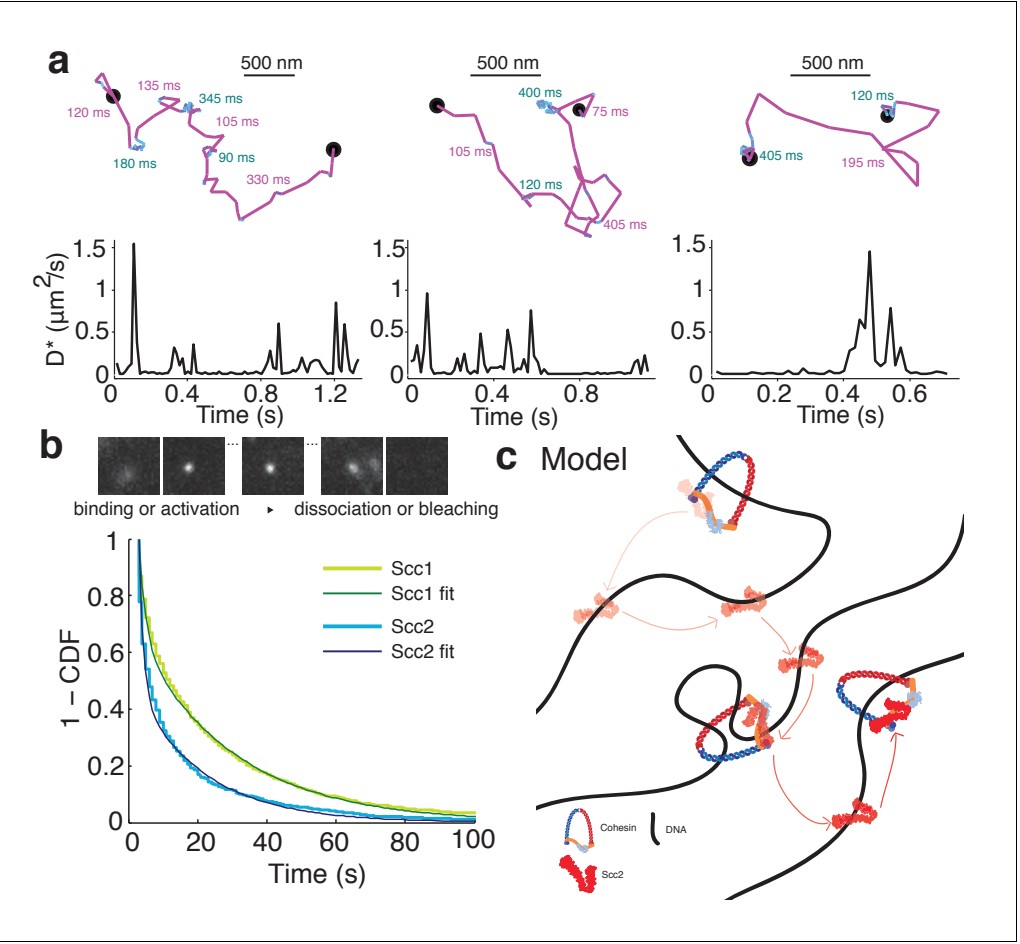

**Figure 7.** Scc2 hops between cohesin binding sites. (**a**) Example tracks show dynamic binding and unbinding of Scc2$^{JF549}$ on a sub-second time-scale. Intervals of diffusive motion (purple) are frequently interrupted by short binding events (cyan). The durations of the mobile or bound intervals are shown. Scale bars = 500 nm. Underneath: Time traces show the instantaneous apparent diffusion coefficient corresponding to each track. (**b**) Binding time of immobile Scc2$^{JF549}$ molecules. Example frames at 1 s exposures showing a blurred diffusing molecule that produces a sharp spot upon binding until it unbinds or bleaches. Distributions (1 - cumulative distribution function) of measured dwell times of immobile Scc2 and Scc1 molecules and fitted curves. n > 10 cells. (**c**) Model of Scc2 dynamics: Scc2 hops between cohesin that is loaded on DNA. Between binding events with cohesin it interacts with chromatin in two binding modes. One is very transient and probably non-specific and the other lasts tens of seconds. The longer interaction may represent Scc2's cohesin-independent role as a transcription regulator.

DOI: https://doi.org/10.7554/eLife.30000.017

region (*Borck et al., 2006*). Why are mice and humans so sensitive to changes in Scc2 expression? We present three observations, which might help answer this question. Scc2 binds to cohesin on DNA after loading, Scc2 is substoichiometric relative to cohesin in wild type cells, and Scc2 rapidly rebinds to cohesin after unbinding. We suggest that the abundance of Scc2 is rate limiting for the ATPase of cohesin that is engaged in loop extrusion. Thus, a lower abundance of Scc2 in CdLS means cohesin is visited less frequently by Scc2 and may reduce the processivity of loop extrusion complexes (*Fudenberg et al., 2016*), and thereby increasing the chance of unregulated enhancer-promoter interactions.

During the course of these studies, we noticed a curious property of Scc2, namely the ability to spread slowly across chromatin. FRAP and tracking of individual Scc2 molecules revealed that Scc2 travels on chromatin by hopping. This feature probably arises because Scc2's association with cohesin has a high on rate and Scc2 is substoichiometric. As a consequence, when Scc2 dissociates from

a cohesin complex it rebinds to one that is in the vicinity before diffusing an appreciable distance across the nucleus. The gradual spreading of Scc2 along chromosomes described here utilised selective photobleaching to create a defined zone or source of labelled Scc2, whose diffusion away from this source was punctuated by repeated dissociation and re-binding events. However, one could imagine situations where specific loci attract large amounts of a protein that then behaves in a manner similar to Scc2 and diffuses gradually away from its source, creating a gradient within surrounding chromatin. We suggest that 'punctuated diffusion' or 'hopping' of this nature could underlie several poorly understood long-range chromosomal regulatory phenomena.

## Materials and methods

### Plasmids

pSpCas9(BB)−2A-Puro (PX459) V2.0 was a gift from Feng Zhang (Addgene plasmid # 62988). The following oligonucleotides were cloned into pX459 at the BbsI restriction sites to make pX459 SCC2 (Hs) 5', pX459 SCC1(Hs) 3' and pX459 WAPL(Hs) M1116 as previously described (*Ran et al., 2013*).

   SCC2(Hs) 5' `TCCAGAAATTCAGGATGAAT`
   SCC1(Hs) 3' `ATAATATGGAACCTTGGTCC`
   WAPL(Hs) M1116 `GCATGCCGGCAAACACATGG`

   A poly Glycine-Serine linker, Blasticidin resistance gene (BSD), GSG-P2A (self cleaving peptide) and the HaloTag were cloned into pUC19 between KpnI and SalI by Gibson Assembly to generate pUC19 NT-BSD-GSG-P2A-HaloTag. If this sequence is inserted after the start codon of a gene equimolar amounts of BSD and N-terminally HaloTagged protein of interest are expressed (*Stewart-Ornstein and Lahav, 2016*). The reverse was also assembled to make pUC19 CT-HaloTag-GSG-P2A-BSD for C-terminal tagging. BSD-GSG-P2A-HaloTag-Linker was cloned between 1 kb sequences homologous to the five prime end of human *NIPBL* to make pUC19 SCC2 NT-BSD-GSG-P2A-HaloTag. HaloTag-GSG-P2A-BSD was cloned between 1 kb sequences homologous to the three prime end of human *SCC1* to make pUC19 SCC1 CT-HaloTag-GSG-P2A-BSD.

### Cell culture

Scc1-AC Tir1 cells were a gift from Masato Kanemaki and cultured as previously described (*Natsume et al., 2016*). HeLa S3 cells were obtained from ATCC (ATCC Cat# CCL-2.2, RRID:CVCL_0058). HeLa Kyoto cells expressing mouse GFP-Scc2 from a stably integrated BAC were a gift from Anthony Hyman and Ina Poser (mouse NIPBL-NFLAP #5701) (*Poser et al., 2008*). Cell lines were tested and confirmed to be mycoplasma-free using MycoAlert Mycoplasma Detection Kit (Lonza, LT07-318). Halo-Scc2 and Scc1-Halo cell lines were generated by cotransfection of the appropriate pX459 and donor vector using TransIT-LT1 (Mirus Bio, MIR 2306). Two days post-transfection cells were plated at low density and blasticidin (Invitrogen, R21001) was added to medium at 5 µg/ml for both HCT116 and HeLa cells. When colonies were clearly visible they were isolated using cloning cylinders and split into two 96-well plates. Homozygous clones were identified by PCR with primers outside the homology arms of the donor plasmid. To deplete Scc1-mClover-mAID, 500 µM auxin sodium salt (Sigma, I5148) was added to the medium two hours before imaging.

### Fluorescent labelling

JF549-HaloTag ligand was a gift from Luke Lavis (*Grimm et al., 2015*). HaloTag labelling was as previously described except 100pM HaloTag-JF549 was used for residence time analysis (*Rhodes et al., 2017*). Anti-Scc1 (Millipore Cat# 05–908, RRID:AB_11214315) was used at 1:100 dilution.

### Confocal microscopy and FRAP

Confocal live-cell imaging was performed on a spinning disk confocal system (PerkinElmer Ultra-VIEW) with an EMCCD (Hamamatsu) mounted on an Olympus IX81 microscope with Olympus 60 × 1.4 N.A. objective. During imaging, cells were maintained at 37°C and 5% $CO_2$ in a humidified chamber.

   For spot FRAP of JF549, ten prebleach images were acquired then a 2.5 µm circle was bleached with the 568 nm laser (75% laser power) and recovery images were acquired. Fluorescence intensity measurements were made using ImageJ. Fluorescence intensity was measured in the bleached and

unbleached regions and a region outside of any cell (background). The background intensity was subtracted from the bleached and unbleached intensities. The relative intensity between bleached and unbleached was calculated by dividing background corrected unbleached intensity by the background corrected bleached intensity. The mean of the relative intensity of prebleach images was calculated and used to normalise all the values so that the relative intensity before bleaching had a mean of 1. The mean normalised relative intensity of all repeats was calculated for each time point and plotted.

## Single-molecule tracking experiments

For single-molecule tracking experiments, we used a custom TIRF/HiLo microscope described in (*Wegel et al., 2016*). Briefly, a fibre-coupled 561 nm laser (Toptica iChrome MLE) was focused into the back focal plane of an Olympus 100x NA1.4 objective. By translating the position of the focus away from the optical axis, we controlled the angle of the excitation beam to maximise the signal-to-noise ratio. JF549 fluorescence was collected by the same objective, split from the excitation beam using a dichroic mirror and emission filter (Chroma), and focused onto an EMCCD camera (Andor iXON 897 Ultra) using a 300 mm tube lens. This resulted in a magnification with pixel size of 96 nm. We used an objective collar heater and heated stage insert to maintain a sample temperature of 37°C during imaging. After identifying an area for imaging, fluorescence was pre-bleached until single molecules were sufficiently sparse for localisation and tracking. For rapid tracking of diffusing and bound molecules, we acquired movies with continuous 561 nm excitation at 50 mW intensity at the fibre output and a frame rate of 64.5 frames/s and exposure time of 15 ms. Each movie typically comprised 5.000 frames and contained several nuclei. For experiments to measure single-molecule binding times, we recorded movies of 300 frames under continuous 1 mW 561 nm excitation and a frame rate of 1 frame/s and exposure time of 1 s.

## Single-molecule tracking analysis

Data analysis was performed in MATLAB (MathWorks) using software that was previously described (*Uphoff et al., 2014*). In each frame, fluorescent molecules were detected based on an intensity threshold, and their localisations determined to 25 nm precision by fitting an elliptical Gaussian Point Spread Function (PSF). Subsequently, localisations that appeared in consecutive frames within a radius of 0.48 μm were linked to tracks. A memory parameter allowed for molecules to be tracked if they blinked or disappeared out of focus for single frames. Tracks with at least four steps were used to compute apparent diffusion coefficients ($D^*$) from the mean-squared displacement (MSD) on a particle by particle basis: $D^*=\text{MSD}/(4\,dt)$, where $dt$ is the time between frames. We classified bound and mobile molecules based on their apparent diffusion coefficient after correcting for the localisation uncertainty of sigma = 25 nm; Dcorrected = MSD/(4 dt) – sigma$^2$/dt. The fraction of bound or diffusing molecules was then estimated from the fraction of tracks that were below or above a threshold of Dcorrected <0.1 μm$^2$/s. Note that $D^*$ represents an 'apparent' diffusion coefficient that is not corrected for certain biases in single-particle tracking experiments, such as motion blurring or diffusion of molecules out of the focal plane. Therefore, we use it only for relative comparisons between experiments, but not as an absolute measure of the diffusion coefficient.

To estimate binding times from experiments at long exposure times, we tracked localisations using a radius of 0.192 μm. To exclude diffusing molecules from the analysis, we filtered tracks based on the apparent diffusion coefficient and the width of the fitted Gaussian function. The lengths of tracks of stationary molecules with diffraction-limited PSF gave the apparent dwell times of chromatin-bound molecules. The binding time constants were obtained by fitting the distribution of dwell times with a double exponential decay function. In order to correct for biases that underestimate the true binding times, we followed the procedure described in the main text and in (*Hansen et al., 2017*; *Uphoff et al., 2013*). Specifically, we used the fact that Scc1 is stably bound (15–30 min) on the time scale of the measurement (<60 s per observed molecule), so that any disappearance of fluorescent Scc1 molecules is due to photobleaching/blinking or movement out of the focal plane. This apparent dwell time was tBleach = 28.3 s. The binding times of Scc2 were then calculated from the fitted dwell time constants using the equation: tBound = tDwell*tBleach / (tBleach - tDwell).

## Acknowledgements

We are very grateful to Masato Kanemaki for sharing the Scc1-AC Tir1 HCT116 cell line. We thank Ina Poser and Tony Hyman for sending the LAP-Scc2 HeLa cells. We are also grateful to Luke Lavis for contributing the JF549 dye. We thank Maurici Brunet-Roig for help with FACS and David Sherratt and Francis Barr for comments on the manuscript.

## Additional information

### Funding

| Funder | Grant reference number | Author |
| --- | --- | --- |
| Wellcome | 091859/Z/10/Z | Kim Nasmyth |
| H2020 European Research Council | 294401 | Kim Nasmyth |
| Cancer Research UK | C573/A12386 | Kim Nasmyth |
| Wellcome | 101636/Z/13/Z | Stephan Uphoff |

The funders had no role in study design, data collection and interpretation, or the decision to submit the work for publication.

### Author contributions

James Rhodes, Conceptualization, Data curation, Formal analysis, Validation, Investigation, Visualization, Methodology, Writing—original draft, Project administration, Writing—review and editing; Davide Mazza, Formal analysis; Kim Nasmyth, Conceptualization, Supervision, Funding acquisition, Writing—original draft, Writing—review and editing; Stephan Uphoff, Conceptualization, Resources, Data curation, Software, Formal analysis, Supervision, Funding acquisition, Validation, Visualization, Methodology, Writing—original draft, Project administration, Writing—review and editing

### Author ORCIDs

James Rhodes http://orcid.org/0000-0001-5853-7343
Davide Mazza http://orcid.org/0000-0003-2776-4142
Kim Nasmyth https://orcid.org/0000-0001-7030-4403
Stephan Uphoff http://orcid.org/0000-0002-3579-0888

### Decision letter and Author response

Decision letter https://doi.org/10.7554/eLife.30000.019
Author response https://doi.org/10.7554/eLife.30000.020

## Additional files

### Supplementary files

• Transparent reporting form
DOI: https://doi.org/10.7554/eLife.30000.018

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
