## [Decision Letter]

Thank you for submitting your article "Scc2/Nipbl hops between chromosomal cohesin rings after loading" for consideration by *eLife*. Your article has been reviewed by three peer reviewers, and the evaluation has been overseen by Andrea Musacchio as the Senior Editor and Reviewing Editor.

The reviewers have discussed the reviews with one another and the Reviewing Editor has drafted this decision to help you prepare a revised submission.

Summary:

The ring-shaped cohesin complex topologically entraps chromosomes and regulates diverse processes, including transcription and sister-chromatid cohesion. The cohesin core consists of the Smc1-Smc3 heterodimer, the Scc1 kleisin subunit that links the ATPase head domains, and a HEAT-repeat-containing protein called SA1/2 in human cells. Cohesin loading to chromosomes requires a loosely bound HEAT-repeat protein called Scc2 (or NIPBL), possibly a modulator of coherin's ATPase activity.

Hitherto, Scc2 was only known to be important in the cohesin loading reaction. This manuscript by Uphoff and colleagues now postulates a role for Scc2/Nipbl following cohesin deposition on chromosomes. Using fluorescence recovery after photobleaching (FRAP) and single-molecule tracking, the authors show that Scc2/Nipbl is generally more dynamic than cohesin and that it exhibits a "hopping" motion suggestive of association with successive chromosome-bound cohesin molecules. These results suggest that it regulates the function of cohesin post-loading. The authors show that Scc2/Nipbl co-localizes with the high-order assembly of cohesin along the chromosome axis (so-called vermicelli) that become evident in Wapl-deficient cells. The authors also show that Scc2/Nipbl dissociates from chromosomes in mitosis in Wapl-deficient cells, despite the fact that cohesin remains bound to mitotic chromosomes in these cells, strongly suggesting that the Scc2-cohesin interaction is weakened in mitosis and possibly explaining the lack of cohesin loading in mitosis.

Understanding the dynamic behavior of cohesin and its loader complex, as well as their interactions, represents an important step forward in identifying the mechanism of cohesin's function. The manuscript represents therefore an important conceptual step forward and is a strong candidate for publication in *eLife*.

Essential revisions:

1) The elegant experimental setup of tracking individual Halo-tagged Scc2 molecules in cultured cells provides indirect evidence that Scc2 interacts with cohesin complexes that had already been loaded onto chromosomes. The authors observe differences in chromatin association dynamics of Scc2 upon deletion of WAPL or upon depletion of the cohesin subunit Scc1. The major conclusion from these experiments is that Scc2, while being able to directly bind chromatin, mainly interacts with chromosome-bound cohesin. However, the authors exclude in their analysis the possibility of an indirect effect on the Scc2 dynamics when chromosome structure is altered by manipulating the levels of chromosome-bound cohesin. For example, Scc2 might simply diffuse more slowly on the densely-packed chromatin structure in 'vermicelli' chromosomes because of a higher local concentration of chromatin binding sites. Cohesin depletion, in contrast, is thought to relax chromatin loops. If Scc2 would track along chromatin fibers, this would provide an alternative explanation why the speed of Scc2 movement would increase if such loops are relaxed by Scc1 depletion. It is therefore possible that the observed differences in the chromatin association dynamics of Scc2 are a consequence of a perturbed interphase chromatin structure.

Providing direct evidence that the "hopping" behavior of Scc2 reflects an interaction with chromosome-bound cohesin would be greatly desirable. For this, the authors may consider directly disturbing the interaction between cohesin and its loader. Specifically, if the authors could engineer a mutant version of Scc2 that fails to interact with cohesin, then it should be possible to express this mutant version as an additional (ectopic) copy and the only fluorescently labelled version of Scc2. Cohesin would most likely be loaded normally by the endogenous Scc2, but the authors would only follow the fluorescently labeled mutant Scc2 in their FRAP experiments. We realise that unless such mutant Scc2 were already available, obtaining one may create a serious bottleneck.

As an alternative, the authors could repeat the single molecule tracking experiments in cells in which Scc2 and Scc1 are simultaneously labeled. Even though double-labelling single-molecule experiments are challenging, they would allow the authors to address whether Scc2 moves between bound cohesin molecules and could hence provide strong support for a "hopping" mechanism. A very similar proposal was raised independently by two reviewers, and is repeated below again as point 3.

2) One issue of the manuscript is that the co-localization of Scc2 with chromosomal cohesin proposed in the authors' model is at odds with ChIP-seq data, which didn't detect such a co-localization at the major cohesin peaks that co-localize with CTCF. To explain this conflict, the authors make the intriguing suggestion that CTCF prevents Scc2 binding at cohesin stalling sites. Assuming that a large fraction of cohesin is associated with CTCF in interphase cells, CTCF depletion should increase the fraction of cohesin-bound Scc2 if the authors' hypothesis were correct. Can this be tested experimentally? Upon CTCF depletion, it might also be possible to co-localize cohesin and Scc2 signals by calibrated ChIP-seq. Similarly, calibrated ChIP-seq might be able to detect Scc2 and cohesin co-localization in WAPL-depleted cells. It might furthermore be possible to test competition between Scc2 and CTCF binding to cohesin biochemically.

3) One point regarding the interpretation of the data should be considered. As written the manuscript gives the impression that Scc2 hops along the chromosomes from one cohesin complex to another. However, given the low stoichiometry of Scc2 compared to Scc1, can the authors rule out the alternative model that a small amount of cellular Scc1 also hops on and off chromosomes together with Scc2? Can the behaviour of Scc2 that is not bound to cohesin be distinguished from the Scc2-cohesin complex in these experiments? In this case it might not be a post-loading function of Scc2, rather a loading-coupled translocation of cohesin that is being visualised. This possibility does not detract from the value of the paper but it would be useful to include an explicit discussion of evidence for and against this point. Although out of the scope of the current study, perhaps single molecule tracking of Scc1 would help address this in the future. Similarly, how far does the behavior of Scc1 mimic that shown for Scc2? The authors cite previously published FRAP data for Scc1 but the experimental set up was different so it is possible that further information could be gained from examining Scc1 and Scc2 side by side.

We realise that the proposed experiments are potentially demanding, and we do not deem them strictly necessary. However, we would be grateful if you could carefully consider the points raised, as the manuscript would benefit significantly if they could be addressed.

---

## [Author Response]

Essential revisions:1) The elegant experimental setup of tracking individual Halo-tagged Scc2 molecules in cultured cells provides indirect evidence that Scc2 interacts with cohesin complexes that had already been loaded onto chromosomes. The authors observe differences in chromatin association dynamics of Scc2 upon deletion of WAPL or upon depletion of the cohesin subunit Scc1. The major conclusion from these experiments is that Scc2, while being able to directly bind chromatin, mainly interacts with chromosome-bound cohesin. However, the authors exclude in their analysis the possibility of an indirect effect on the Scc2 dynamics when chromosome structure is altered by manipulating the levels of chromosome-bound cohesin. For example, Scc2 might simply diffuse more slowly on the densely-packed chromatin structure in 'vermicelli' chromosomes because of a higher local concentration of chromatin binding sites. Cohesin depletion, in contrast, is thought to relax chromatin loops. If Scc2 would track along chromatin fibers, this would provide an alternative explanation why the speed of Scc2 movement would increase if such loops are relaxed by Scc1 depletion. It is therefore possible that the observed differences in the chromatin association dynamics of Scc2 are a consequence of a perturbed interphase chromatin structure.

Single-molecule tracking was able to address this possibility, as described in the manuscript:

“Interestingly, although the relative abundance of unbound Scc2 molecules was reduced by Wapl deletion and increased by cohesin degradation, the apparent diffusion coefficients of unbound Scc2 molecules remained unchanged in both situations. This indicates that the movement of Scc2 during the search for binding sites is not affected by the presence or absence of cohesin or by the reorganisation and compaction of chromatin caused by Wapl deletion.”

In addition, Scc2 colocalises with cohesin in Wapl∆ vermicelli. If Scc2 diffused predominantly on chromatin it should colocalise better with DNA, which is mostly found outside vermicelli.

We do not claim that chromatin has no role in Scc2 binding and movement. On the contrary, we think that Scc2 also binds chromatin sites independent of Scc1 (e.g. “after degrading Scc1 by auxin treatment, Scc2 still displayed transient binding events as seen in untreated cells. Therefore, we interpret these binding events on a time-scale of ~100 ms as cohesin-independent chromatin interactions during the target search.”).

Providing direct evidence that the "hopping" behavior of Scc2 reflects an interaction with chromosome-bound cohesin would be greatly desirable. For this, the authors may consider directly disturbing the interaction between cohesin and its loader. Specifically, if the authors could engineer a mutant version of Scc2 that fails to interact with cohesin, then it should be possible to express this mutant version as an additional (ectopic) copy and the only fluorescently labelled version of Scc2. Cohesin would most likely be loaded normally by the endogenous Scc2, but the authors would only follow the fluorescently labeled mutant Scc2 in their FRAP experiments. We realise that unless such mutant Scc2 were already available, obtaining one may create a serious bottleneck.

This would be an extremely interesting experiment and while Scc2 mutants defective in Scc1 binding have been identified, these have only been experimentally verified in vitro with yeast proteins^1^. Several of these mutations are viable in yeast and some occur naturally in certain species indicating that they are not truly defective in Scc1 binding. In short, we believe that generating and characterising these mutations and performing the experiments would not be achievable in an appropriate timeframe.

As an alternative, the authors could repeat the single molecule tracking experiments in cells in which Scc2 and Scc1 are simultaneously labeled. Even though double-labelling single-molecule experiments are challenging, they would allow the authors to address whether Scc2 moves between bound cohesin molecules and could hence provide strong support for a "hopping" mechanism. A very similar proposal was raised independently by two reviewers, and is repeated below again as point 3.

By nature of the single-molecule tracking technique, only a very small fraction of molecules resides in the fluorescent state at any time. The unobserved non-fluorescent molecules are still performing their functions and changing their positions in the cell. Therefore, it would be extremely rare to simultaneously observe 2 different molecules both in the fluorescent state during a transient interaction. For Scc2 and cohesin, this would be entirely impossible for three reasons:

1) Because there are so many molecules of cohesin bound to DNA it would be impossible to identify the positions of a sufficiently high fraction of Scc1 molecules before the cell/chromatin has moved.

2) The density of Scc1 molecules is so high that it would be difficult to resolve them individually and to determine which Scc1 position an individual Scc2 is bound to.

3) The short-lived hopping we observe by particle tracking reflects non-specific DNA interactions independent of cohesin (as discussed in the text). The cohesin interactions last ~60 seconds, which is far longer than the observation window of single molecules during the fast tracking experiments (fractions of a second).

2) One issue of the manuscript is that the co-localization of Scc2 with chromosomal cohesin proposed in the authors' model is at odds with ChIP-seq data, which didn't detect such a co-localization at the major cohesin peaks that co-localize with CTCF. To explain this conflict, the authors make the intriguing suggestion that CTCF prevents Scc2 binding at cohesin stalling sites. Assuming that a large fraction of cohesin is associated with CTCF in interphase cells, CTCF depletion should increase the fraction of cohesin-bound Scc2 if the authors' hypothesis were correct. Can this be tested experimentally?

We believe this experiment will be impossible to perform in the allotted time and the proposed experiment would not necessarily answer the question.

1) We think that in an individual cell most cohesin is not found at CTCF sites. Cohesin colocalises with CTCF in ChIP-seq analyses because they are focused exclusively on peaks, which are aggregates of millions of cells. We expect that most cohesin is in transit to CTCF sites. Therefore, inactivation of CTCF may not have a strong effect on Scc2 dynamics in individual cells.

2) The cell line to perform this experiment would take several months to generate. This is because we would have to knock in the mAID tag into CTCF and HaloTag into Scc2 in the Tir1 expressing cell line. It is not possible to start with the existing Halo-Scc2 cell line as that would require introducing Tir1.

Upon CTCF depletion, it might also be possible to co-localize cohesin and Scc2 signals by calibrated ChIP-seq. Similarly, calibrated ChIP-seq might be able to detect Scc2 and cohesin co-localization in WAPL-depleted cells.

This is an excellent idea for an experiment and we are hoping to do this. Establishing calibrated ChIP-seq with the HaloTag is ongoing in Professor Nasmyth’s lab however we have run into several problems. We will not be able to include it in this manuscript, as it is very unlikely to be working in the coming weeks.

It might furthermore be possible to test competition between Scc2 and CTCF binding to cohesin biochemically.

Again, this is a very good idea. At present we do not have human cohesin, Scc2 or CTCF purification up and running. This would be a separate project in itself but is well worth considering.

3) One point regarding the interpretation of the data should be considered. As written the manuscript gives the impression that Scc2 hops along the chromosomes from one cohesin complex to another. However, given the low stoichiometry of Scc2 compared to Scc1, can the authors rule out the alternative model that a small amount of cellular Scc1 also hops on and off chromosomes together with Scc2? Can the behaviour of Scc2 that is not bound to cohesin be distinguished from the Scc2-cohesin complex in these experiments? In this case it might not be a post-loading function of Scc2, rather a loading-coupled translocation of cohesin that is being visualised. This possibility does not detract from the value of the paper but it would be useful to include an explicit discussion of evidence for and against this point.

We do not think that Scc2 hopping can be explained by cohesin hopping on chromatin in complex with Scc2 for the following reasons:

1) It is indeed possible to distinguish between Scc2 and the putative Scc2-cohesin complex, as the latter would show a lower diffusion coefficient than Scc2 alone due to the large size of the complex. We found that the diffusion coefficient of the mobile Scc2 molecules was unaffected by Scc1 depletion. If Scc2 diffused in complex with cohesin the diffusion coefficient should be increased by depletion of the Scc1 subunit.

2) Similarly, there was no change in the diffusion coefficient of mobile Scc2 molecules in wapl∆ where there are no freely diffusing cohesin molecules.

3) In wapl∆ Scc2 is hopping on vermicelli. If Scc1 were hopping on chromatin Scc2 should be found equally distributed on chromatin, not where the stable cohesin is found.

Although out of the scope of the current study, perhaps single molecule tracking of Scc1 would help address this in the future. Similarly, how far does the behavior of Scc1 mimic that shown for Scc2? The authors cite previously published FRAP data for Scc1 but the experimental set up was different so it is possible that further information could be gained from examining Scc1 and Scc2 side by side.

We had already performed single-molecule tracking of Scc1 (see Figure 5, and text).

We realise that the proposed experiments are potentially demanding, and we do not deem them strictly necessary. However, we would be grateful if you could carefully consider the points raised, as the manuscript would benefit significantly if they could be addressed.

The suggestions made by the reviewers are good ones, which would certainly yield important discoveries about the nature of the interaction between Scc2 and cohesin. However, as the reviewers pointed out, these experiments are demanding and several are projects in their own right. Furthermore, we believe that the observations presented in this manuscript stand well by themselves and the conclusions are well supported.

1) Kikuchi, S., Borek, D. M., Otwinowski, Z., Tomchick, D. R. & Yu, H. Crystal structure of the cohesin loader Scc2 and insight into cohesinopathy. Proc. Natl. Acad. Sci. U.S.A. 113, 12444–12449 (2016).